# Hard Flaccid Syndrome: A Biopsychosocial Management Approach with Emphasis on Pain Management, Exercise Therapy and Education

**DOI:** 10.3390/healthcare11202793

**Published:** 2023-10-22

**Authors:** Evdokia Billis, Stavros Kontogiannis, Spyridon Tsounakos, Eleni Konstantinidou, Konstantinos Giannitsas

**Affiliations:** 1Laboratory of Clinical Physiotherapy and Research, Department of Physiotherapy, School of Health Rehabilitation Sciences, University of Patras, 26504 Rio, Greece; 2Department of Urology, University Hospital of Patras, 26504 Rio, Greece; 3Department of Psychiatry, University Hospital of Patras, 26504 Rio, Greece; 4Pelvic Floor Physiotherapy, 54622 Thessaloniki, Greece

**Keywords:** hard flaccid syndrome, biopsychosocial, exercise therapy, education, chronic pelvic pain, pain management, pelvic floor exercises, graded exposure, stretches

## Abstract

Hard flaccid syndrome (HFS) is a rather rare, acquired clinical entity affecting young men’s well-being, sexual and social life. HFS presents with a cluster of symptoms including penile-specific somatosensory disturbances, a semi-rigid penis at the flaccid state without any stimulation or desire, erectile dysfunction, perineal and/or penile pain, associated urinary symptoms, emotional distress as well as other psychosocial and stress-related manifestations. Although its pathophysiology is still not well understood, initial penile trauma causing minor nerve and vascular disturbances to the penis and associated pelvic floor musculature is suggested to trigger the syndrome. Despite the scarcity of research on HFS, the present report describes a case of a young male clinically diagnosed with HFS, who benefited from a biopsychosocial management strategy, focusing on pain management, therapeutic exercise approaches, such as pelvic floor exercise re-education, graded exposure to activity as well as education on lifestyle and stress-related modifications. This holistic management approach has been clinically reasoned in this case report, and the need for more evidence-based studies developing diagnosing criteria, elaborating pathophysiological mechanisms and testing the efficiency of different therapeutic options is highlighted.

## 1. Introduction

Hard flaccid syndrome (HFS) is a rather rare, acquired clinical entity affecting young men and is characterized by a cluster of penile-specific somatosensory disturbances, affecting men’s sexual and social life as well as couples’ well-being. In particular, the symptomatology of HFS includes a semi-rigid penis at flaccid state without any stimulation or desire, penile sensory changes (such as coldness, numbness, dysesthesia, etc.), perineal and/or penile pain, erectile dysfunction (such as decreased frequency of erections, reduced libido, etc.) and emotional distress, usually presented with anxiety, depression, sleep disturbances as well as a general feeling of a lack of relaxation in the perineal/penile area [1,2]. Associated urinary symptoms, such as painful urination or weak urinary flow have also been reported [3].

Although pathophysiological mechanisms are not well defined or understood, symptom onset is suggested to be related to penile trauma via incident(s) of vigorous sexual intercourse or masturbation, including jelqing (a set of penis stretching exercises that some believe can enlarge the size of the penis) among other activities, traumatizing the base of the erect penis and its associated neurovascular structures, which provide nerve and blood supply to the pelvic floor muscles and the penis itself; thus, explaining the somatosensory changes described by patients in this area [1,4]. These presentations in turn, cause emotional distress and/or anxiety, initiating a sympathetic reaction, which is suggested to induce a series of adverse effects, such as prolonged pelvic floor muscle spasms, further obstructing the skin, musculature and neurovascular supply to this area.

As the literature is scarce concerning HFS, various conservative treatment options have been suggested, relying on each patient’s evaluation findings, including stress and behavioural modification management approaches, pelvic floor muscle relaxation, biofeedback, low-intensity shockwave treatment, analgesics for pain, and phosphodiesterase-5 inhibitors for the erectile dysfunction among other treatments, etc. [1,2,3]. However, in view of the limited evidence concerning this syndrome, no definite conclusions on the most effective treatment strategies can be made.

The present report describes a case of a young male clinically diagnosed with HFS, who benefited from a biopsychosocial approach, focusing on pain management, graded exposure to activity, exercise therapy-specific treatment approaches including pelvic floor exercise re-education, manual therapy-assisted techniques, etc., as well as education on lifestyle and stress-related modifications.

## 2. Materials and Methods

### 2.1. Case Presentation

The patient was a 30-year-old who visited one of the university hospitals in mainland Greece. He was a private employee, having a relatively demanding, sedentary-based job. The subject gave his informed consent for being included in the case study.

Upon urological examination performed by the hospital’s urologist, the patient had a semi-rigid penis without sexual stimulation or desire. In addition, he also had erectile dysfunction and mild lower urinary tract symptoms for the last 6 months. His triggering episode appeared to be an intense sexual intercourse approximately 6 months ago, as the next day he started developing the painful symptoms in the base of his penis and the perineum, together with reported changes in the shape of his penis and a ‘stiffening feeling’ around his superficial pelvic floor muscles. Concerning his sexual symptoms, the patient had an International Index of Erectile Function Score (IIEF)-5 of 18, indicating mild erectile dysfunction [5]. His sexual desire was low, and his ejaculation/orgasm were normal. With self-stimulation, he reported decreased penile rigidity but with the ability to penetrate. The random erections were diminished, and sexual performance anxiety was high. Concerning his urinary symptoms, the patient was feeling some pain (moderate in nature) during urination, as well as a feeling of incomplete bladder emptying and increased frequency of urination at times when his symptoms were exacerbated. Since the beginning of his symptoms, he became very anxious and worried about his problem and interrupted any sexual activity or other intense physical activity or sport.

Upon physical examination, the urologist did not detect any abnormality with the penis or testicles. There was no medical history of medications or drug/alcohol abuse. His past medical and drug history were clear. His previous surgical history entailed a minor surgery for a short frenulum at the age of 17. His urine lab tests were negative, the duplex colour ultrasound of the penile arteries revealed normal parameters, no arteriovenous fistulae were noticed, and his metabolic and hormonal status were also normal. Tadalafil 5 mg daily was initiated for the erectile dysfunction.

He was then referred to a psychiatrist for evaluation. The qualified psychiatrist did not diagnose any major psychiatric illness, other than anxiety issues, mainly due to mistaken cognitive beliefs. On the Hospital Anxiety and Depression Scale (HADS) he scored 11 for anxiety (HADS-A) and 9 for depression (HADS-D), indicating moderate levels of anxiety and marginally elevated depression levels [6,7,8]. At that time, he was suggested to proceed with psychotherapy treatment, but the patient declined due to lack of time.

The patient was then referred for physiotherapy by the urologist (who was the initial healthcare contact for the patient), with the clinical diagnosis of hard flaccid syndrome. The urologist suggested a physiotherapy (PT) assessment of his pelvic floor muscles and subsequent PT management (if needed). The patient presented to the PT clinic with a persistent history of deep perineal pain (5–6/10 on VAS) with moderate sensory disturbances (mainly coldness and dysesthesia) around the superficial perineal and deeper pelvic floor area. The pain seemed to be aggravated when standing. He also reported some mild (but controlled) constipation. He had tried several (mostly internet-based) self-management strategies by then, such as superficial perineal massage, reverse Kegel exercises, plant-based analgesic medications, avoidance of any sexual activity, etc., without much effect.

Upon clinical PT assessment, the patient was thin, with good but slightly sway back posture. Concerning palpation of his abdomen, he presented with tightness across the right abdominal musculature (especially rectus abdominus, external oblique and iliopsoas) without tender or painful points around his abdominal muscles, pubic symphysis or other anterior pelvic areas. Concerning palpation of his perineum (externally), he reported tenderness and tightness more so in the right area of the transverse perineal muscle. Tightness was also evident upon palpation over the obturator foramen area (more evident again in the right side). Upon internal (digital) examination, his posterior pelvic floor muscle area was tight, with local pain reproducing his pain state. Two active trigger points (TPs), approx. 4 and 7 o’clock of the pelvic ring, referring pain to the perineum were found, which were compatible with Anderson et al.’s [9] male trigger point evaluation findings. These TPs were possibly indicative of some tightness in the puborectalis, pubococcygeus and ischiococcygeus muscles. However, his pelvic floor muscles presented with discrete contraction and relaxation. According to the pelvic floor muscle assessment conducted manually through the Perfect Scheme approach [10], pelvic floor muscle strength was good, 4+ on the 5-point modified Oxford scale, and the pelvic floor achieved elevation during contraction. Duration of the contractions was short (3–4 s), with 6 repetitions being enough before muscle fatigue and a reduction in strength was reached, indicating a tendency for muscle fatigue. Ten fast contractions were well conducted (though muscle relaxation over the later repetitions was reduced).

Upon real-time transabdominal ultrasound, performed by the PT for rehabilitative purposes and based on Khorasani et al.’s study protocol [11], pelvic floor muscle mobility was low, which is often visible in pelvic pain syndrome presentations (though no gold standard or cutoff values are presently established for bladder base movement on ultrasound).

### 2.2. Clinical Reasoning and Goal Setting Management

The present report collates clinical presentations seen on males presenting with HFS [2]. Although current evidence on conservative management is still scarce, treatment has been organized on three therapeutic axes, based on assessment findings and supporting a biopsychosocial approach:*Bio*—pelvic floor re-education/improvement of pelvic floor musculature. Based on the clinical and ultrasound assessment findings, pelvic floor muscles required improvement in extensibility, stamina and needed de-activation of the active TPs. Most of these methods have previously been indicated in relevant assessment findings for HFS [1,2,3] as well as other pelvic floor-related dysfunctions [9,12,13,14], aiming to improve muscle physiology and better pelvic floor function.*Psycho*—pain management and coping strategies. Based on the subject’s painful symptoms and considering the difficulties in coping with his health issue, an educative approach including pain neuroscience education principles, such as explaining pain, pain physiology, how thought and cognitive processes affect pain and discussing different day-to-day management strategies focusing on his work, social and sexual life were utilized [15,16,17,18]. In addition, a graded strategy to activity exposure was employed, as indicated in the literature, to eliminate negative pain behaviours [19].*Psychosocial*—lifestyle and stress-release modifications. Physical therapeutic strategies aiming to dampen sympathetic activity and stress, such as deep (diaphragmatic) breathing exercises, bladder training strategies, ergonomic modifications, stretching, and relaxation approaches were also deemed appropriate to control the stress-related parameters and improve urination quality [3,9,20]. Pacing and graded exposure strategies to activity were also utilized to improve patient’s function, reduce fear or pain and anxiety [19].

## 3. Results

### Subsequent Physiotherapy Visits

Overall, the patient undertook five physiotherapy visits within a 3-month interval. Based on the aforementioned goal-setting programme, the management details are summarized in Table 1.

Gradual improvement in all clinical, stress-related, social and sexual manifestations of his problem were reported following this physiotherapy course. At the end of the fifth visit, the patient reported more than 85% improvement. Somatosensory disturbances were vaguely evident and only at times where his stress levels were elevated. Pain in the perineum/base of the penis and the stiffening penile feeling were absent. Urination and constipation issues were all normalized. HADS-A and HADS-D were reduced to 7 and 6, respectively. Regarding clinical evaluation, trigger points were absent and pelvic floor performance on the Perfect Scheme was improved. Ultrasound imaging of pelvic floor muscle mobility was also improved from all positions (supine, sitting and standing). His IIEF-5 score increased to 22 (normal erectile function), his morning erections started to appear more frequently, he reported 9/10 erection rigidity in self-stimulation and the urologist suggested tapering down tadalafil 5 mg from daily to every other day.

The following two issues were still evident at the end of the fifth PT visit: his stress-related responses and his sexual function. Firstly, he acknowledged the tremendous impact that stress had in triggering his symptoms; he therefore started focusing more on stress-relieving techniques and coping methods related to stress-induced parameters (such as his work). Secondly, his sexual activity; although he had initiated sexual intercourse on the last month with good feedback from the patient in terms of penile rigidity, erection function, satisfaction, libido, he was still fearful on re-occurrence or flare-ups. Thus, the sexual performance anxiety was still high, and this perhaps had to be further addressed.

## 4. Discussion

This case report presents a multimodal management approach for HFS. Although diagnostic and pathophysiological mechanisms are still not fully acknowledged, understood, or established [2,25], and despite the scarcity of research in HFS, the clinical presentation and overall symptomatology of the patient were in accordance with current published literature on the syndrome [1,2,3,4,26]. In addition, the patient’s clinical presentation revealed similarities with various other pelvic syndromes and chronic pain states, which do not affect only the somatosensory and motor components of the patient, but also the psychological, social, relationship and sexual dimensions of the patient; thus, requiring a biopsychosocial management approach. However, more reports are suggested to be made public to establish the full clinical entity of HFS; certainly, a medical consensus is urgently needed in order to determine clear diagnostic criteria for HFS.

The therapeutic management approach undertaken, which was based on a holistic strategy, utilizing three therapeutic axes (pelvic floor muscle management, pain management and lifestyle/stress-related modifications) followed a biopsychosocial management approach, and it seemed to interrupt the vicious cycle of symptoms being described [1,2]. Indeed, such an approach is compatible with current evidence on various pelvic syndromes and chronic pain states [2,15,16,17,27]. Although the HFS-specific outcomes and overall quantitative outcomes are limited, the aforementioned approach proved favourable for the patient. Apart from the pelvic floor muscle rehabilitation undertaken as well as the soft tissue management approaches on the abdominal and pelvic areas [13,14,28], education on pain physiology and behaviour and how this can affect stress, fear, thought processing and centrally sensitize pain is an effective approach, which over recent years has been used for chronic pelvic pain [15,17]. In particular, individually-tailored neurophysiological pain education has been reported to reduce fear and avoidance as well as catastrophizing and worrying thoughts, leading to reduced anxiety about returning to functionality [15,17,23]. Furthermore, graded exposure to activity, following careful monitoring of activity and identification of the active components of the patient’s pain manifestations have been recommended as successful techniques for changing pain behaviour and related fear or activity avoidance in several chronic musculoskeletal pain states [19]. In conjunction with the above, application of stress-related techniques, such as diaphragmatic breathing, advice on activity pacing and graded exposure to patient’s ‘fearful’ activities also proved beneficial. Such approaches, although limited in evidence are often used and suggested in painful and stressful pelvic pain dysfunctions [9,20,24].

However, stress and sexual life did not fully improve. This may partly be attributed to the rehabilitation time, as 3 months may not be enough for psychological, lifestyle and behavioural changes to take place, and partly because more specific management strategies might be required. For example, Edwards et al. [18] proposed a more extensive interdisciplinary approach for patients with persistent pain and associated pelvic pain problems for specifically improving their sexual life. It may be that such a model would prove more appropriate at this stage, or that, a more specific stress-management approach by other skilled health professionals, such as clinical psychologists or psychiatrists, would further enhance the patient’s stress management [16].

Nevertheless, this report’s strength, apart from utilizing a multimodal biopsychosocial approach, is that it provides substantial detail in the rehabilitation management undertaken (Table 1). This may be helpful in guiding healthcare professionals’ rehabilitation management when dealing with such a syndrome. However, further work is necessary to determine other parameters, such as frequency and length of therapy, and percentage of therapy input on each therapeutic domain (biomedical, psychological and social).

As this is a case report, no inferences or generations can be made. In view of the limited research for this rare syndrome, there is a need for more evidence-based studies developing a series of clinical directions for HFS, including diagnosing criteria (as there is still controversy on its clinical diagnosis), aetio-pathophysiological mechanisms involved in its development, evaluating the efficacy of different therapeutic modalities as well as achieving interprofessional consensus for an established management approach. Along with these, syndrome-specific outcome measures are urgently needed to be developed, to elaborate treatment efficacy [26].

## 5. Conclusions

In this clinical case study diagnosed with hard flaccid syndrome, the value of applying a biopsychosocial approach has been highlighted. Apart from the initial medical and psychosocial approach, new advances within physiotherapy rehabilitation, focusing on a tailor-based pelvic floor muscle re-education exercise programme in combination with graded activity exposure, pain management, education, stress and lifestyle modification approaches, has been clinically reasoned, resulting in favourable treatment outcomes for the patient’s pain, associated clinical symptoms, social, sexual and functional status.

## Figures and Tables

**Table 1 healthcare-11-02793-t001:** Details of the physiotherapy management program.

Therapeutic Axis/Treatment Goal	Techniques	Parameters/Examples
***Bio***— Pelvic floor re-education—Improvement of pelvic floor musculature/improve extensibility, muscle stamina, control [10,11,12,13,14]	Pelvic floor muscle stretching exercises	Taught and performed by the subject independently daily, e.g., manual stretches of ischial tuberosities in full hip flexion in lying position, deep squatting poses with legs wide apart, frog-type stretches, etc.
	Pelvic floor muscle endurance exercises	Discreet submaximal contractions followed by relaxation with gradually increasing timings, e.g., started with 6 reps of 4 s contraction holds and 4 s relaxation holds, with progression every 2 weeks with increased holds for 2 more seconds and 1–2 more reps (build up to 10).
	Trigger point de-activation (ischaemic deep pressure massage via internal manual approach)	TP massage treatment of puborectalis, pubococcygeus and ischiococcygeus muscles/ischaemic pressure with pressure application of approx. 1 min and repeat until reduction in pain and/or tension over TP is achieved (required in three out of the five PT visits)
	Deep and superficial front fascial line massage [21,22]	Instrumented soft tissue mobilization and deep frontal line massage (targeting iliopsoas, rectus abdominus, obliques muscles) and whole anterior body stretching exercises in supine and standing positions (required in three visits).
***Psycho***— Pain management and coping strategies approach [15,17,18,19,23]	Pain neuroscience education	Education on explaining pain, pain physiology and how cognition, thoughts, beliefs, etc., cause pain sensitization.
	Discussion of different management strategies focusing on graded exposure and activity pacing to work, social and sexual life	Graded exposure and pacing on specific day-to-day activities following careful feedback from patient’s weekly activity diary and motivational interview strategies on promoting functionality.
***Psychoscocial***— Lifestyle and stress-release modifications/reduce sympathetic activity and stress [1,2,3,19]	Deep (diaphragmatic) breathing exercises [9,20,24]	Taught and performed by the subject independently on a daily basis from supine (at home) and sitting positions (at work, toileting, etc.). At least 2 × 20 min daily and when else needed. Progression with 0.5 kg weight on diaphragm (supine position) and gradually decreasing breathing session time.
	Bladder training strategies	Take time, ensure full relaxation and good positioning prior to urination and complete urination with a strong pelvic floor contraction.
	Ergonomic advice, activity pacing and other relaxation approaches	Taught good sitting posture, disrupting prolonged standing/sitting with small breaks of stretching/other activities, alternating sitting in an office chair with sitting in a large and softer physiotherapy ball, breaking down activity into smaller parts, etc.

## Data Availability

No new data were created.

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
