# Peer review of "Hard Flaccid Syndrome: A Biopsychosocial Management Approach with Emphasis on Pain Management, Exercise Therapy and Education"

_healthcare, 2023, doi:10.3390/healthcare11202793_

Round 1

Reviewer 1 Report

Thank you for the opportunity to review the manuscript entitled Hard flaccid syndrome: a biopsychosocial management approach with emphasis on pain management, exercise therapy and education.” This is an interesting case, with a novel treatment. However, some recommendations on the presentation of the case are provided below:

1.       Avoid to refer to the patient as Mr X; this is only used once and has no purpose. Refer to the patient rather.

2.       Did the physiotherapist do a urological examination, sexual dysfunction, ED (IIEF) assessment and LUTS assessment? Please more clearly indicate what specialists did the different components: e.g. urogenital history, examination biochemical assessments (urinary) and diagnosis was made by an appropriate specialist (the urologist mentioned prior to the referral in Ln67 is presumed at the moment). This also includes ultrasound of penis and prescription for ED, which I also presume is not done after referral to the physiotherapist, but prior to referral.  

3.       Likewise, please clearly show that qualified psychiatrists did the psychiatric evaluation

4.       The case presentation starts with the referral to physiotherapy (Ln67), but then there is referral to physiotherapy later in Ln 101; This relates to point 2 and 3 above, and needs to be more clearly presented please at each stage, and which specialists conducted different aspects of this case report

5.       Ln104 – 106: This appears more case history, and should be part of the case history; unless this is additional information obtained at the physio rather than the urologist? It is unclear

6.       The discussion should make use of the case report by Nico et al. (2022): https://www.sciencedirect.com/science/article/abs/pii/S174360952200251X

7.       Include discussion on strengths and limitations of this case report, beyond this being only a case report. This can relate to diagnostic criteria (or lack thereof), validated diagnostic tools, ability to subjectively and/or objectively determine impact of therapy, length of therapy, and so on.

8.       The conclusion is too long, and part of it is really discussion. The conclusion should have 2 – 3 take away points relevant to the case, which is about the treatment approach and not so much on the diagnostics, mechanisms, efficacy evaluation, etc.

Some minor language editing can be considered 

Author Response

Dear Editor in Chief and Reviewers,

On behalf of all the authors, thank you very much for your time spent in reviewing our work as well as your constructive comments re. our (submitted to Healthcare Special Issue) case report titled: “Hard flaccid syndrome: a biopsychosocial management a-proach with emphasis on pain management, exercise therapy and education”. We have tried to address all comments. Please, find a point-by-point reply to every comment. All changes are being highlighted in the revised manuscript, which has now been much improved.

Thanking you in advance

Evdokia Billis (corresponding author)

Reviewer 1.

  1. Avoid to refer to the patient as Mr X; this is only used once and has no purpose. Refer to the patient rather.

We have amended this, thank you

  1. Did the physiotherapist do a urological examination, sexual dysfunction, ED (IIEF) assessment and LUTS assessment? Please more clearly indicate what specialists did the different components: e.g. urogenital history, examination biochemical assessments (urinary) and diagnosis was made by an appropriate specialist (the urologist mentioned prior to the referral in Ln67 is presumed at the moment). This also includes ultrasound of penis and prescription for ED, which I also presume is not done after referral to the physiotherapist, but prior to referral.  

We have amended these parts, summarizing who did what (lines 71, 87, 94, 100-102 & 131). Hope it reads clearly now (thank you)

  1. Likewise, please clearly show that qualified psychiatrists did the psychiatric evaluation

We have added this as well - thank you (line 94)

  1. The case presentation starts with the referral to physiotherapy (Ln67), but then there is referral to physiotherapy later in Ln 101; This relates to point 2 and 3 above, and needs to be more clearly presented please at each stage, and which specialists conducted different aspects of this case report

Thank you for your comment. Yes, you are right about not being clear at which stage was the patient referred for physiotherapy. We have erased referral to physiotherapy in line 67 and have amended both sections (see highlighted sentences). Hope it reads well now.

  1. Ln104 – 106: This appears more case history, and should be part of the case history; unless this is additional information obtained at the physio rather than the urologist? It is unclear

Yes, thank you again. As this (lines 104-106) was part pf the physiotherapy assessment, we have made this clearer (by adding ‘The patient presented to the PT clinic with ….’, line 103). Hope it is clear now.

  1. The discussion should make use of the case report by Nico et al. (2022): https://www.sciencedirect.com/science/article/abs/pii/S174360952200251X

We were not aware of this case report, which actually strengthens our clinical reasoning and overall decision making. We have now included it in our report (from clinical reasoning section onwards), along with another one, just published report, which is also insightful  https://pubmed.ncbi.nlm.nih.gov/37670085/  (thank you)

  1. Include discussion on strengths and limitations of this case report, beyond this being only a case report. This can relate to diagnostic criteria (or lack thereof), validated diagnostic tools, ability to subjectively and/or objectively determine impact of therapy, length of therapy, and so on.

We have now re-organised the final two paragraphs of the discussion (lines 230-242) according to your suggestions (strengths, limitations and suggestions based on this case report) -thank you

  1. The conclusion is too long, and part of it is really discussion. The conclusion should have 2 – 3 take away points relevant to the case, which is about the treatment approach and not so much on the diagnostics, mechanisms, efficacy evaluation, etc.

We have reduced this section (from 13 to 7 lines), focusing on the treatment approach, as suggested.

Reviewer 2 Report

Thank you for inviting me to review this interesting paper. This case report titled “Hard flaccid syndrome: a biopsychosocial management approach with emphasis on pain management, exercise therapy and education” presents an interesting treatment approach to managing hard flaccid syndrome.

The case report is well-structured, written and referenced. It provides a detailed approach to the management of hard flaccid syndrome. In my opinion, it is useful for physicians and physiotherapists. In many cases, in diseases with a mental component where stress, anxiety and depressive symptoms play a role, management is particularly challenging. For this reason, a combination of many modes of treatment is a good choice. Here the multifaced approach includes pain management, exercises (including pelvic floor), lifestyle interventions and reduction in stress.

I would only advise to do some minor language changes. For example in line 67, a case report is usually started with “A 30-year-old man referred by his urologist…”

Was it important that he was a private employee? And what does it mean? Perhaps it would be better to specify what type of work he was doing, eg. physical or sitting office work instead of indicating the public or private sector. But I leave it to your discretion.

The term “an automatic erection” is usually linked with erections triggered by alprostadil. Is this the desired meaning? Perhaps you mean random erections?

Finally, please follow the rules for using abbreviations in scientific literature. Abbreviations should be defined at first mention and used consistently thereafter (improper use of abbreviations is in line 91 - ultrasound (US)). Abbreviations can be introduced when used at least 3 times.

Please follow the rules for using abbreviations in scientific literature. Abbreviations should be defined at first mention and used consistently thereafter (improper use of abbreviations is in line 91 - ultrasound (US)). Abbreviations can be introduced when used at least 3 times.

Author Response

Dear Editor in Chief and Reviewers,

On behalf of all the authors, thank you very much for your time spent in reviewing our work as well as your constructive comments re. our (submitted to Healthcare Special Issue) case report titled: “Hard flaccid syndrome: a biopsychosocial management a-proach with emphasis on pain management, exercise therapy and education”. We have tried to address all comments. Please, find a point-by-point reply to every comment. All changes are being highlighted in the revised manuscript, which has now been much improved.

Thanking you in advance

Evdokia Billis (corresponding author)

The case report is well-structured, written and referenced. It provides a detailed approach to the management of hard flaccid syndrome. In my opinion, it is useful for physicians and physiotherapists. In many cases, in diseases with a mental component where stress, anxiety and depressive symptoms play a role, management is particularly challenging. For this reason, a combination of many modes of treatment is a good choice. Here the multifaced approach includes pain management, exercises (including pelvic floor), lifestyle interventions and reduction in stress.

Thank you very much.

I would only advise to do some minor language changes. For example in line 67, a case report is usually started with “A 30-year-old man referred by his urologist…”

Thank you for your comment. We have checked the whole text again for language issues. Re. line 67, as Reviewer 1 suggested to mention referral to the urologist in further down on the text (lines 100-102), this section has been changed. Hope this reads well now.

Was it important that he was a private employee? And what does it mean? Perhaps it would be better to specify what type of work he was doing, eg. physical or sitting office work instead of indicating the public or private sector. But I leave it to your discretion.

We have added this information (line 68). The patient’s job was mainly sedentary, which seemed to affect him (more so, however, in terms of stress rather than movement-wise). Interestingly, within Greece, private-sector jobs are considered highly ‘unstable’ and are ultimately potential ‘stressors’.

The term “an automatic erection” is usually linked with erections triggered by alprostadil. Is this the desired meaning? Perhaps you mean random erections?

Thank you for your comment. Yes we meant random erections (we have replaced the word  ‘automatic’ with the word ‘random’)

Finally, please follow the rules for using abbreviations in scientific literature. Abbreviations should be defined at first mention and used consistently thereafter (improper use of abbreviations is in line 91 - ultrasound (US)). Abbreviations can be introduced when used at least 3 times.

We have amended this throughout the text. Thank you very much